# Effect of using home-based dynamic intermittent pneumatic compression therapy during periods of physical activity on functional and vascular health outcomes in chronic stroke: A randomized controlled clinical trial

**James Faulkner**[ID][1]*, **Eloise Paine**[2], **Nick Hudson**[2], **Scott Hannah**[2], **Amy Dennis-Jones**[3ᵒ],
**Louis Martinelli**[4ᵒ], **Helen Hobbs**[4]

1 Primary Care Research Centre, Faculty of Medicine, University of Southampton, Southampton, United Kingdom, 2 Department of Sport, Allied Health Professions and Social Work, University of Winchester, Winchester, United Kingdom, 3 Hobbs Rehabilitation, Bristol, United Kingdom, 4 Hobbs Rehabilitation, Winchester, United Kingdom

ᵒ These authors contributed equally to this work.
* J.A.Faulkner@soton.ac.uk

## Abstract

### Background

Intermittent pneumatic compression (IPC) therapy may benefit stroke patients by eliciting more intensive training sessions that may result in better health, mobility and ultimately quality of life. The purpose of this randomized controlled trial was to assess the effect of using a home-based IPC device on functional outcomes and vascular health in individuals with chronic stroke.

### Methods

Thirty-one stroke survivors (64.3±14.3y; 4.3±2.7y since stroke) completed pre- and post-intervention assessments of functional capacity (six-minute walk test [6MWT], timed-up-and-go, 10m walk test), vascular health (pulse wave analysis, carotid-femoral pulse wave velocity), and physical activity. Following the pre-assessment, individuals were randomly assigned to either a daily, 12-week, home-based IPC group, or to a usual care control (CON) group. Outcomes were assessed using analysis of covariance (ANCOVA), controlling for age and any baseline differences.

### Results

Following ANCOVA, a significant increase in 6MWT walking distance was observed post-assessment for the IPC (Mean ± SD [95%CI]; 188 ± 19 m [177–199m]) but not the CON group (167 ± 19 m [157–178m]) (p < 0.05). A significant reduction in peripheral

**Data availability statement:** All relevant data are within the paper and its Supporting Information files.

**Funding:** JF Received funding for the research Grant number: N/A The full name of each funder: WinBack Medical URI: https://winback-medical.com/ Did the sponsors or funders play any role in the study design, data collection and analysis, decision to publish, or preparation of the manuscript?: NO.

**Competing interests:** The authors have declared that no competing interests exist.

**Abbreviations:** ADL, Activities of daily living; AIx, Augmentation Index; cfPWV, Carotid-femoral pulse wave velocity; CON, Control; cSBP, Central systolic blood pressure; DVT, Deep vein thrombosis; IPC, Intermittent pneumatic compression; PWA, Pulse wave analysis; RPE, Ratings of perceived exertion; SBP, Systolic blood pressure (peripheral); TUG, Timed up-and-go; UK, United Kingdom

systolic blood pressure was reported at the post-assessment for the IPC group (136.2 ± 8.0 mmHg [131.9–140.4 mmHg]) but not for CON (142.2 ± 8.0 mmHg [138.1–144.6 mmHg]) ($p < 0.05$). Similar findings were observed for central systolic blood pressure. Physical activity levels significantly increased at the post-assessment for IPC (1857 ± 879 MET·min$^{-1}$·week$^{-1}$ [1390–2325 MET·min$^{-1}$·week$^{-1}$]) but not for the CON group (1161 ± 879 MET·min$^{-1}$·week$^{-1}$ [677–1645 MET·min$^{-1}$·week$^{-1}$]), while for time spent sitting, a significantly greater reduction was observed at the post-assessment for the IPC group (396 ± 86 mins [350–442 mins]) compared to CON (486 ± 86 mins [439–534 mins]) (both $p < 0.05$).

## Conclusions

The observed improvements in functional mobility, cardiovascular health, increased physical activity and reduced sedentary time demonstrates important clinical implications of 'home-based' IPC therapy as a clinical training aid for stroke rehabilitation. Home-based IPC therapy could serve as an adjunct to conventional rehabilitation, however, further research is needed to determine whether IPC therapy can sustain or improve function over time for individuals in the chronic stage of recovery.

## Introduction

Stroke is the second-leading cause of death globally, the third-leading cause of death and disability combined [1], and causes more permanent disability than any other neurological condition [2]. In the United Kingdom (UK) there are over one million individuals living with the symptoms of stroke [3]. With stroke incidence expected to increase by 60% by 2035 [4], and thus, higher associated financial and societal costs [4], identifying and optimizing safe, efficient, and effective treatment plans for stroke survivors is essential [5].

Initiating effective rehabilitation interventions soon after a stroke onset can significantly augment the recovery process and reduce functional disability. However, despite diligent rehabilitation efforts, a substantial proportion of stroke survivors (~ 50 to 60%) contend with varying degrees of motor impairment [6], with about half of these survivors requiring assistance during activities of daily living [7]. Moreover, a large proportion (80%) of stroke survivors experience gait impairments [7], adversely affecting their independence, physical activity levels, and overall quality of life [8,9]. Integrating repetitive, task-specific training that includes actual walking practice has shown promise in improving mobility and balance among stroke survivors [10], increasing physical activity engagement and thereby enhancing quality of life.

Evidence suggests that people with chronic stroke may experience reduced blood flow and decreased arterial diameter in the hemiparetic limb [11]. These arterial changes may influence exercise performance and functional ambulation, thus making physical activity or activities of daily living even harder to complete [11]. Increasing physical activity can be an effective intervention for improving blood flow delivery in the hemiparetic limb [11]. Treatment therapies implemented within a 'home-based' environment could be highly efficacious as it may allow physiotherapists to implement rehabilitation activities without being physically present. This may not only increase the potential volume of (physical) activity but may also contribute to the formation of habits within a familiar context that leads to long-term behavior change [12]. For example, a home-based overground robotic-assisted gait training program improved functional (e.g., 6-minute walk test [6MWT], balance) [13] and vascular [14] health outcomes

in chronic stroke patients on completion of the 10-week intervention, and the benefits were sustained for a further 3 months.

Compression therapy is a rehabilitation tool where pressure is applied to the body to activate blood circulation, and venous and lymphatic return [15]. The physiological, physical, neuromuscular, biomechanical, and/or perceptual benefits of compression therapy on performance and recovery has been widely researched in an athlete-based setting [16,17]. Intermittent pneumatic compression (IPC) devices are used within a clinical rehabilitation setting, such as for preventing deep vein thrombosis (DVT) post-surgery or for those with venous insufficiency [15]. IPC devices used within an acute stroke setting can significantly reduce the risk of DVT [18]. There may also be benefits of using IPC therapy for chronic stroke patients, as if venous return can be increased, individuals may be able to engage in more physical activity, and more intensive training sessions [19], resulting in better health, mobility and ultimately quality of life [20].

The purpose of this study was to assess the effect of using a home-based IPC device on functional outcomes, including the 6MWT (primary outcome), and vascular health in individuals with chronic stroke. It was hypothesized that individuals taking part in a 12-week home-based IPC training intervention would improve functional (e.g., 6MWT) and vascular outcomes.

## Methods

This study was a dual-center (University of Winchester, UK; Hobbs Rehabilitation, UK), parallel-group, randomized controlled clinical trial (Fig 1). The study protocol received institutional human research ethics approval from the University of Winchester's Faculty of Health and Wellbeing ethics committee (approval code: RKEEC210801_Faulkner) and was registered with Clinical Trials.gov Protocol Registration and Results System (NCT05276453; https://clinicaltrials.gov/ct2/show/NCT05276453). Intermittent pneumatic compression therapy required

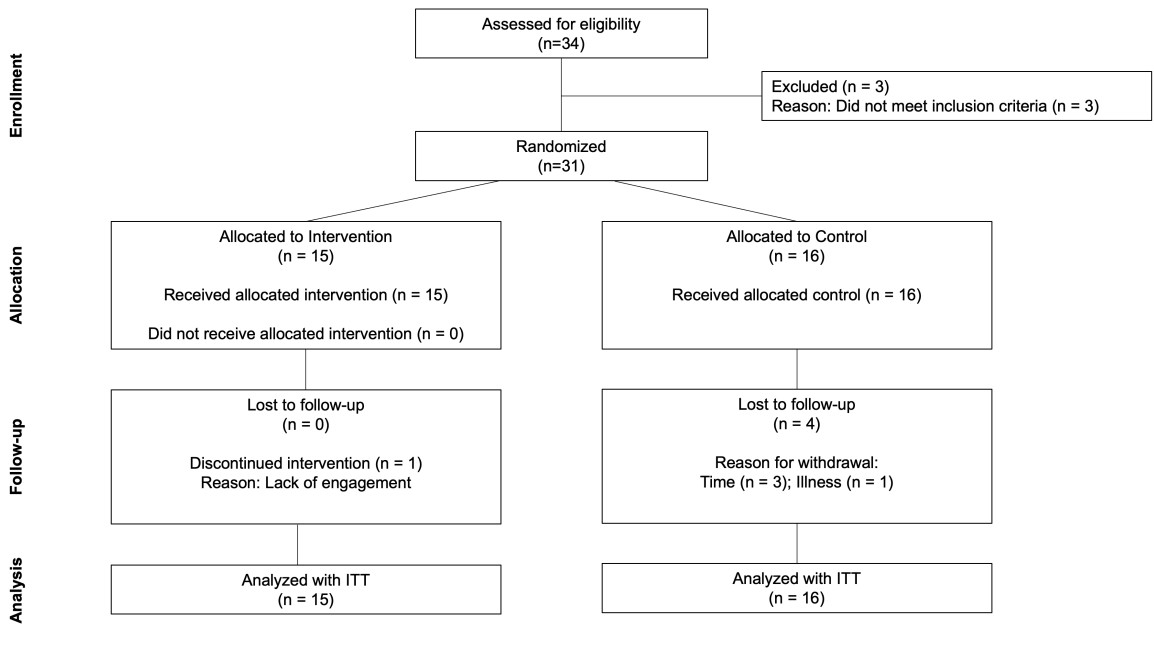

Note: ITT, Intention-to-treat analysis

**Fig 1. Participant recruitment and retention.**

participants to use the GMOVE Suit (Winback Medical, Nice, France), an active, lower-limb, pressotherapy system. GMOVE Suits were provided freely by Winback Medical who had no input or influence on the data analysis or manuscript preparation. Baseline assessments took place in early April 2022. The final follow-up assessment was undertaken in July 2022. Based on the findings of Ivey et al. [21] and when using a two-tailed 5% significance level and a power of 80%, a sample size of 15 per group was calculated to detect a mean difference of 32m (pooled SD; 45m) for the 6MWT between groups and a moderate-to-large effect size (cohen $d$ = 0.7).

## Participants

Participants were diagnosed with stroke by a UK National Health Service (NHS) stroke consultant. Participants were recruited from Hobbs Rehabilitation, a UK-based neuro-physiotherapy practice, following completion of all recommended inpatient and outpatient NHS care [22]. Prior to the study's initiation, written informed consent was obtained for all participants.

Participants meeting the inclusion criteria were, at the time of enrollment, between 3 months and 7 years post-stroke, residing in the community, medically stable, cognitively capable, and able to stand and step with assistance or aid, as indicated by a Functional Ambulation Category between 2 and 5 [23]. Exclusion criteria included unstable cardiovascular conditions, unresolved deep vein thrombosis, severe osteoporosis, recent fractures of the symptomatic limb, individuals unable to bear weight, and open wounds.

## Experimental design

Participants took part in four primary testing sessions. This included two baseline (Pre) testing sessions conducted at the Exercise Physiology laboratory at the University of Winchester (Part 1) and Hobbs Rehabilitation (Part 2; Fig 2), and two identical post-intervention assessments at the same locations. For laboratory visits, participants were asked to refrain from food for 3 hours, caffeine for 12 hours, and intense physical activity for 24 hours before the assessments. At the pre-assessment, and following written informed consent, participants completed a health history questionnaire. Participants' height (Seca, 213, Germany) and weight (Seca, Quadra 808, Germany) were then measured using a stadiometer and electronic scales, respectively. Following 15 minutes of supine rest whereby resting blood glucose and total cholesterol were obtained (CardioCheck, Fitech, UK), regional measures of arterial stiffness as determined by carotid-femoral pulse wave velocity (cfPWV), and pulse wave analysis (PWA), were assessed to determine participants' vascular health. Other measures of vascular health were also captured, including brachial-femoral and femoral-anterior tibial PWV, however, these measures will be reported in a separate manuscript as they are exploring some methodological considerations associated with assessing PWV, in comparison to the widely used and cited cfPWV. Participants then completed 6MWT [24] and Timed Up-and-Go (TUG) [25] functional assessments. To minimize the effect of fatigue on the data collected, participants were asked to attend Hobbs Rehabilitation for further outcome assessments, approximately 7 days after the initial laboratory visit. Here, trained physiotherapists with more than 15 years of practical experience working with stroke patients assessed the following: Functional Ambulation Classification [23], Berg Balance Scale [26], Fugl Meyer Assessment [27], and Activities Balance Confidence (ABC) Scale [28]. Participants also completed a 10m walk test [29], Sit-to-stand assessment [30], the International Physical Activity Questionnaire short-form [IPAQ-SF]) [31] and the SF-12 as a measure of quality of life [32]. On completion of the assessment, participants were randomized to either a 12-week home-based IPC therapy group or to a 'usual care' control (CON) group, using a computer-based random number generator operated by a researcher external to the study. Block randomization was employed to ensure

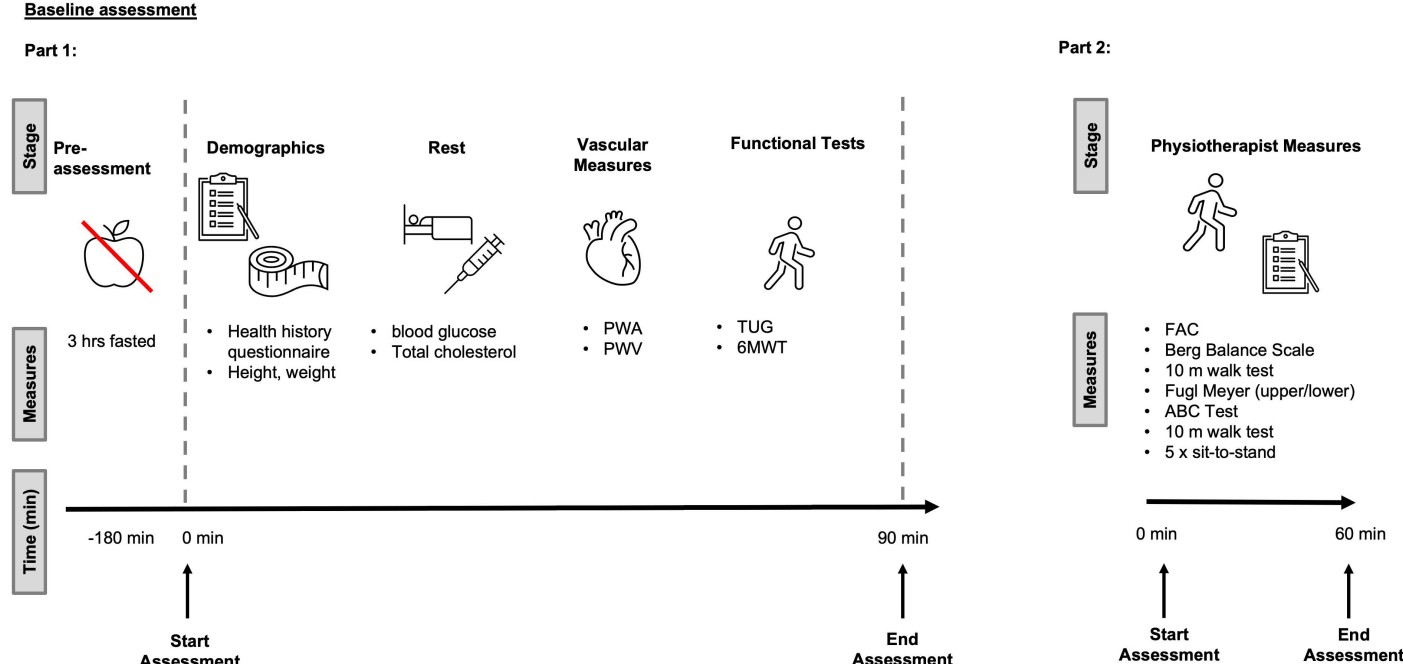

*Note:* ABC, Activities Balance Confidence; FAC, Functional ambulation classification; min, minutes; PWA, Pulse wave analysis; PWV, Pulse wave velocity; TUG, Timed up-and-go test; 6MWT, Six minute walk test.

**Fig 2. Schematic of assessments conducted during part 1 and part 2 of the Pre-assessments.**

an equal distribution of participants across the groups. The study used eight blocks, each consisting of four participants, two assigned to the IPC group and two to the Control group, to meet the required sample size. Within each block, participants were randomized to either the IPC or control group. Identical assessments were completed post-intervention. Participants and therapists administering the IPC intervention were not blinded to group allocation. Outcome assessors and data analysts were blinded from group allocation.

**IPC Intervention.** Participants in the IPC group were familiarized with the lower-limb compression device (Fig 3) by a specialist neuro-physiotherapist with experience of using the device. The therapist demonstrated and provided step-by-step instructions for how to wear and set up the device, and participants were provided a range of exercises (e.g., walking, sit-to-stand, step-ups) that they were encouraged to undertake within the 12-week training program. Within this session, a starting compression pressure was also identified (usually between 50–70 mmHg). Participants were encouraged to wear the device for a minimum of 30 minutes each day. The first IPC therapy session took place at Hobbs Rehabilitation with a physiotherapist present. This session was designed to ensure participants were confident in wearing the IPC device and in undertaking the prescribed physical activities. Thereafter, participants had face-to-face or telephone-based discussions with a therapist every other week (weeks 3, 5, 7, 9, 11), to ensure participants were: i) using the IPC device at the most appropriate yet comfortable compression pressure settings, and ii) to troubleshoot any issues with the device. Throughout the 12-week program, participants were advised to exercise at standardized ratings of perceived exertion (RPE) using the Borg 6–20 RPE scale [33]: RPE 11 (week 1 to 3), RPE 13 (week 4 to 6), RPE 15 (week 7 to 9) and RPE 17 (week 10–12). Participants recorded the frequency, duration, and intensity (e.g., RPE) of wearing the IPC

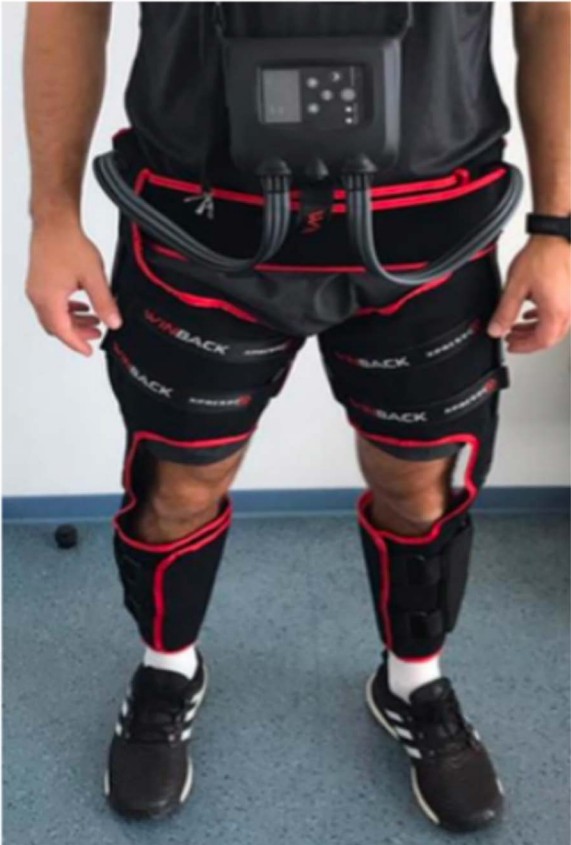

**Fig 3. Image of the intermittent pneumatic compression device (GMove Suit).**

device in a home-based setting in an exercise diary, as well as any additional physical activities or other usual daily activities which they were encouraged to engage with (e.g., physiotherapy sessions). Study-related and non-study-related serious adverse event (SAE) and adverse event (AE) tracking occurred throughout the study. SAE and AEs were recorded in the exercise diary and were discussed with therapists during their face-to-face or telephone-based discussions.

**Control group.** If participants were randomized to the CON group, they were encouraged to engage in a minimum of 30 minutes of daily physical activity (e.g., walking, sit-to-stand, step-ups), similar to the IPC group, and to continue with other usual daily activities (e.g., physiotherapy sessions). Participants were contacted fortnightly by telephone by the therapists involved in the study to 'check in' on them. Participants were also provided an activity diary to record the frequency, duration, and intensity of the activities that they participated in throughout the 12-week intervention.

## Outcome measures

**Functional outcome measures.** All functional outcome measures were undertaken by physical trainers who had > 10 years of experience in administering these assessments to neurological populations.

*Six-minute walk test (6MWT):* The 6MWT was the primary outcome measure of the study as it provides an overall measure of an individual's walking ability, indicates physical

incapacity, and is sensitive to change as a result of rehabilitation therapy which targets walking performance [24]. The 6MWT was conducted indoors on a flat walkway. Participants were required to walk between two cones 10m apart for a total of six minutes and were instructed to complete as far a distance as possible.

*Timed Up-and-Go test (TUG):* Participants were asked to start in a seated position on a chair placing their back against the back of the chair and arms resting on their thighs. If an assistive device was required for walking this was placed nearby. Following a countdown, the participants stood up, walked to a cone that was 3m away, walked around the cone and back to the chair and sat back down. Participants were asked to complete this test as quickly but as safely as they could. The time to complete the test was recorded using a stopwatch. This test was repeated three times, and participants were given up to ~2 minutes rest between each trial if needed. The fastest trial was used in the analysis.

*10m walk test:* Participants were asked to walk as fast as they could between two cones situated 10 m apart. The time to complete the test was recorded using a stopwatch. Three trials were performed, and the fastest trial was analyzed.

*Sit-to-stand test:* Participants completed five continuous, timed, sit-to-stand exercises. Participants started seated, with their feet on the floor, shoulder width apart and their back resting on the back of a chair. Following a countdown, participants stood up straight, then sat back down on the chair, and repeated this movement five times.

**Vascular outcome measures.** *Pulse wave velocity (PWV):* Arterial pulse transit time was determined using the SphygmoCor XCEL (AtCor Medical, XCEL, Australia), a device that facilitates the simultaneous assessment of proximal and distal arterial waveforms. In our study, a tonometer was placed on the left carotid artery (proximal site) while a

volume-displacement oscillometric cuff was wrapped around the left thigh at the level of the femoral artery (distal site). To calculate carotid-femoral pulse wave velocity (cfPWV), the carotid-femoral pulse transit time is divided by the arterial path length. Carotid–femoral pulse transit time is based upon the duration between the diastolic feet of the proximal and distal arterial pulse waveforms [34]. To estimate the arterial path length (carotid-femoral distance) the linear distance from the suprasternal notch to the top of the cuff at the leg's centerline was measured, then subtracted from the distance from the suprasternal notch to the left carotid artery measurement site. Three cfPWV measurements were taken, and an average was derived from the two closest readings.

*Pulse wave analysis (PWA):* To conduct PWA, one investigator used the SphygmoCor XCEL to record oscillometric pressure waveforms on the left upper arm, adhering to manufacturer recommendations [35]. In each measurement cycle, a 60-second recording of brachial blood pressure was followed by a 10-second sub-systolic recording. A validated transfer function was utilized to generate a corresponding aortic pressure waveform [38], enabling the derivation of central systolic blood pressure (cSBP), augmentation index (AIx), and augmentation pressure. The AIx was further adjusted to a heart rate of 75 bpm (AIx75). Three measurements were collected, and the average of the two closest values was computed. Additionally, peripheral systolic and diastolic blood pressures (SBP, DBP), and mean arterial pressure (MAP) were evaluated. Measurements were performed at heart level to mitigate the impact of arm angle variations on AIx.

**Other outcome measures.** *Physical activity:* The IPAQ-SF is a reliable (rho = 0.77–1.00) and valid (r = 0.67) instrument for assessing physical activity (PA) across three distinct categories: walking (3.3 MET), moderate-intensity (4.0 MET), and vigorous-intensity activities (8.0 MET), in addition to tracking average daily sitting time [31]. PA quantification involves calculating activity levels within each category and aggregating them to generate a comprehensive PA score (MET·min$^{-1}$·week$^{-1}$). In this study, participants completed the IPAQ-SF during the pre-assessment and during the final week of the 12-week intervention period.

*Berg Balance Scale (BBS):* The BBS is used to objectively determine a patient's ability (or inability) to safely balance during 14 predetermined tasks. Each task consists of a five-point ordinal scale ranging from 0 to 4, with 0 indicating the lowest level of function and 4 the highest level of function. A score of 56 represents functional balance, while a score ≤ 49 indicates a risk of falls in individuals with stroke [36].

*Activities Balance Confidence (ABC) scale:* The ABC scale is a 16-item self-report measure in which patients rate their balance confidence when performing various ambulatory activities. Each item is rated from 0–100, with an average ABC score reported.

*Fugl-Meyer assessment:* The Fugl-Meyer is designed to assess motor functioning, balance, sensation, and joint functioning in patients with post-stroke hemiplegia. Scoring is based on direct observation of performance by a physiotherapist with experience in using the measure. Scale items are scored on a three-point scale where 0 is the individual cannot perform, one is when the individual can partially perform the movement, and two represents full movement. For this study, upper and lower limb motor function is reported.

*SF-12:* The SF-12 is a 12-item questionnaire that is a widely used measure to assess health-related quality of life in patients with stroke. It provides questions that relate to eight specific domains: Vitality, Global Health, Role Physical, Physical Functioning, Bodily Pain, Role Emotional, Social Functioning, and Mental Health. The SF-12 also derives a psychometrically based Physical and Mental Health Component Score.

On completion of all outcome measures, participants were randomly assigned using a computerized random number generator to either the IPC or CON groups.

## Follow-up

All outcome assessments which were undertaken within the pre-assessments were repeated at follow-up (post-assessments).

## Data analysis

This study is reported in accordance with Consolidated Standards of Reporting Trials (CONSORT) guidelines [37]. Firstly, a series of independent samples t-tests (i.e., age, time since stroke, FAC) or chi-square tests (e.g., sex) were used to compare demographic characteristics and clinical outcomes (e.g., PWA, cfPWV, 6MWT) between groups (ICP, CON) during the pre-assessments. Average daily wear time (in minutes) and the average number of days participants wore the IPC suit is presented at the start (week 1) and end (week 12) of the IPC training program and are presented as mean and standard deviation (SD). Although participants were encouraged to exercise at set RPE's throughout the program, the mean (± SD) RPE participants reported within weeks 1 and 12 are also reported.

To assess the effect of the IPC therapy on functional, vascular, physical activity and quality of life health outcomes, and following checks for normality to determine the data were parametric (e.g., skewness, kurtosis, Shapiro-Wilk test of normality, histograms), a series of mixed model, two-factor mixed model analysis of covariance; Condition (IPC vs. CON) by Time (pre- vs. post-intervention), adjusted for baseline measures and age, were used for all outcome measures. For cfPWV, mean arterial pressure (MAP) was included as an additional covariate as it has been shown to influence arterial stiffness measurements [38]. An intention-to-treat analysis was applied to all repeated-measures procedures, with participants analyzed as randomized, using the last available data to replace any subsequent missing assessments. Partial eta squared ($\eta p^2$) was used to interpret the measure of effect, with the following thresholds used to aid the interpretation: small (.0099), moderate (.0588) and large (0.1379) [39]. Pearson correlation coefficients and Spearman's *rho* were used to explore whether: i. changes in

average daily IPC wear time over the course of the IPC intervention-, and ii. changes in total physical activity as determined by the IPAQ, were correlated with changes in the 6MWT and pertinent blood pressure markers. Statistical analyses were performed using Statistical Package for Social Sciences version 26 (SPSS, Inc., Chicago, IL, USA). Alpha was set at 0.05. All data are reported as means (SD) unless otherwise specified.

## Results

Participant recruitment and retention are presented in Fig 1. End of study was determined due to achieving the desired sample size. The 31 participants who enrolled in the study (64.3 ± 14.3 years) had been living with stroke for ~4 years (Table 1). Physical activity and sitting time were statistically similar between IPC (1427 ± 1033 MET·min$^{-1}$·week$^{-1}$; 451 ± 185 minutes, respectively) and CON groups (1367 ± 1752 MET·min$^{-1}$·week$^{-1}$; 489 ± 201 minutes, respectively) prior to randomization (p > 0.05). Most participants in both IPC and CON groups met recommended physical activity levels (14 out of 15 for IPC; 13 out of 16 for CON) prior to study participation.

One IPC participant terminated their involvement in the study four weeks after study allocation due to a lack of engagement with the IPC device. There were no "probably study-related" SAEs, but there was one "non-study related" SAE (seizure). There were 14 total study-related AEs which included 10 reported pressure discomforts, two minor skin abrasions and two musculoskeletal concerns (e.g., muscle tightness after IPC intervention). All study-related AEs were resolved, and participants continued in the study. There were no falls during the IPC intervention.

For IPC participants, there was an increase in the daily average wear time of the IPC device (30 ± 15 min to 36 ± 12 mins) and RPE (12.0 ± 1.3 to 14.6 ± 3.4), from the first to the last week of the 12-week intervention, respectively. The number of days wearing the IPC suit also increased from the first (4.1 ± 2.1 days) to last week of the program (5 ± 1.9 days).

**Table 1.  Participants demographics (mean ± SD) for IPC and CON at pre- and post-assessment.**

|  | IPC (n = 15) | | Control (n = 16) | |
| --- | --- | --- | --- | --- |
|  | Pre | Post | Pre | Post |
| Age (y) | 64.7 ± 16.4 |  | 67.3 ± 15.0 |  |
| Age range (y) | 32–90 |  | 46–88 |  |
| Sex Male n (%) | 11 (73%) |  | 12 (75%) |  |
| Female n (%) | 4 (27%) |  | 4 (25%) |  |
| Time since stroke (y) | 3.1 ± 2.2 |  | 3.2 ± 1.9 |  |
| FAC | 3.3 ± 0.9 |  | 3.1 ± 1 |  |
| Height (m) | 1.73 ± 0.1 |  | 1.74 ± 0.12 |  |
| Weight (kg) | 82.0 ± 12.4 | 82.6 ± 13.3 | 77.8 ± 14.4 | 78.1 ± 14.5 |
| BMI (kg/m²) | 27.6 ± 3.9 | 27.8 ± 4.2 | 25.4 ± 2.7 | 25.5 ± 2.8 |
| Glucose (mmol/L) | 5.62 ± 1.12 | 6.02 ± 1.22 | 5.35 ± 1.38 | 5.81 ± 1.58 |
| Cholesterol (mmol/L) | 4.83 ± 1.21 | 4.49 ± 1.00 | 4.57 ± 1.09 | 4.59 ± 1.28 |
| Waist (cm) | 94.5 ± 10.1 | 97.0 ± 12.6 | 92.7 ± 10.8 | 94.2 ± 10.9 |
| Hip (cm) | 105.6 ± 5.8 | 105.4 ± 56.0 | 100.3 ± 7.5 | 101.6 ± 7.3 |
| WHR | 0.93 ± 0.09 | 0.92 ± 0.10 | 0.92 ± 0.08 | 0.93 ± 0.08 |

*Note:* CON, Control; FAC, Functional Ambulation Classification; IPC, Intermittent pneumatic compression; WHR, waist-to-hip ratio.

## Functional outcome measures

S1 Table demonstrates the pre- and post-assessment data for all functional outcomes. ANCOVA demonstrated a significant Time by Condition interaction for 6MWT performance ($p < 0.05$, $\eta p^2 = 0.185$). Participants significantly increased their walking distance between pre- (165m) and post- assessments for the IPC group (Mean ± SD [95%CI]; 187 ± 20 m [175–197m]) but not for CON (168 ± 20 m [157–179m]); Fig 4). There were no significant Time by Condition interactions for the terminal RPE reported on completion of the 6MWT, TUG, 10m walk, sit-to-stand, Fugl-Meyer [Upper & Lower] score, Berg Balance Scale [BBS] or ABC scale (all $p > 0.05$). A Time main effect was observed for the ABC scale, with a significant increase in balance confidence observed between pre- (57.6 ± 16.9%) and post-assessment (63.4 ± 18.0%) ($p < 0.05$). For the SF-12, there was no Condition by Time interactions or main effects for the physical or mental component scores (both $p > 0.05$).

## Vascular outcome measures

There were no differences between Conditions (IPC, control) at the pre-intervention assessment for all vascular outcomes (S2 Table). ANCOVA demonstrated a significant Time by Condition interaction for peripheral SBP ($p < 0.05$; $\eta p^2 = 0.137$) and cSBP ($p < 0.05$; $\eta p^2 = 0.149$; Table 2). Significantly greater reductions in pre-assessment peripheral SBP (143.1 mmHg) were observed at the post-assessment for the IPC group (Mean ± SD [95%CI]; 135.8 ± 8.5 mmHg [131.3–140.3 mmHg]) compared to CON (Mean ± SD [95%CI]; 142.4 ± 8.5 mmHg [138.0–146.7 mmHg]). For cSBP, significantly greater reductions in pre-assessment cSBP (130.9 mmHg) were also observed at the post-assessment for the IPC group (Mean ± SD [95%CI]; 124.4 ± 7.4 mmHg [120.5–128.3 mmHg]) compared to CON (Mean ± SD [95%CI]; 130.4 ± 7.4 mmHg [126.6–134.2 mmHg]). There were no other Condition by Time interactions or main effects for all other vascular outcomes ($p > 0.05$; Table 2).

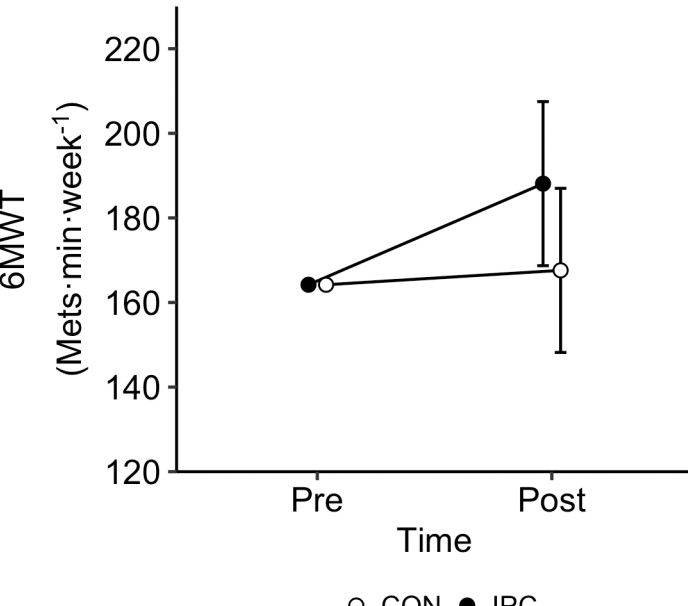

**Fig 4. Mean ( ± SD) 6MWT distance for IPC and CON at pre- and post-intervention assessments, when controlling for differences at baseline.**

**Table 2. Mean (± SD) vascular outcomes for IPC (n = 15) and CON (n = 16) groups at pre- and post-assessment following ANCOVA, including 95% Confidence Intervals and effect sizes ($\eta p^2$).**

| Outcome | Condition | Pre# | Post | 95% CI | $\eta p^2$ |
|---|---|---|---|---|---|
| cfPWV (m/s) | IPC | 8.69 | 8.34 ± 0.90 | 7.82. 8.86 | 0.067 |
| | CON | 8.69 | 8.77 ± 0.90 | 8.28, 9.27 | |
| SBP (mmHg) | IPC | 143.1 | 135.8 ± 8.5* | 131.3, 140.3 | 0.137 |
| | CON | 143.1 | 142.4 ± 8.5 | 138.0, 146.7 | |
| DBP (mmHg) | IPC | 84.6 | 81.2 ± 8.2 | 76.7, 84.6 | 0.009 |
| | CON | 84.6 | 79.7 ± 8.2 | 75.4, 83.9 | |
| PP (mmHg) | IPC | 58.1 | 56.0 ± 9.6 | 50.9, 61.1 | 0.070 |
| | CON | 58.1 | 61.1 ± 9.6 | 56.1, 66.0 | |
| cSBP (mmHg) | IPC | 130.9 | 124.4 ± 7.4* | 120.5, 128.3 | 0.149 |
| | CON | 130.9 | 130.4 ± 7.4 | 126.6, 134.2 | |
| cDBP (mmHg) | IPC | 85.4 | 82.1 ± 4.8 | 79.6, 84.6 | 0.010 |
| | CON | 85.4 | 83.0 ± 4.8 | 80.5, 85.5 | |
| AIx (%) | IPC | 30.5 | 30.3 ± 6.1 | 27.1, 33.6 | 0.014 |
| | CON | 30.5 | 29.0 ± 6.1 | 25.7, 32.2 | |
| AIx75 (%) | IPC | 25.8 | 26.9 ± 6.0 | 23.8, 30.2 | 0.091 |
| | CON | 25.8 | 23.4 ± 6.0 | 20.4, 26.5 | |
| MAP (mmHg) | IPC | 102.7 | 98.6 ± 5.9 | 95.5, 101.7 | 0.030 |
| | CON | 102.7 | 100.5 ± 5.9 | 97.5, 103.5 | |
| HR (bpm) | IPC | 66.2 | 67.2 ± 6.3 | 63.9, 70.6 | 0.001 |
| | CON | 66.2 | 66.9 ± 6.3 | 63.7, 70.2 | |

*Note:* AIx, Augmentation index; AIx75, Augmentation index @ 75 bpm; cDBP, Central diastolic blood pressure; cfPWV, Carotid-femoral pulse wave analysis; CON, Control; cSBP, Central systolic blood pressure; DBP, peripheral diastolic blood pressure; IPC, Intermittent pneumatic compression; HR, Heart rate; MAP, Mean arterial pressure; PP, Pulse pressure; SBP, peripheral systolic blood pressure.

#All pre data has been adjusted for baseline covariates.

*Significant Time by Condition interaction (p < 0.05).

## Physical activity

ANCOVA demonstrated a significant Time by Condition interaction for average weekly physical activity (p < 0.05; $\eta p^2$ = 0.145; Fig 5a) and time spent sitting (p < 0.05; $\eta p^2$ = 0.222; Fig 5b). Significantly greater increases from the pre-assessment physical activity (1399 MET·min$^{-1}$·week$^{-1}$) were observed at the post-assessment for the IPC group (Mean ± SD [95%CI]; 1917 ± 879 MET·min$^{-1}$·week$^{-1}$ [1445–2378 MET·min$^{-1}$·week$^{-1}$]) compared to CON (Mean ± SD [95%CI]; 1226 ± 879 MET·min$^{-1}$·week$^{-1}$ [724–1727 MET·min$^{-1}$·week$^{-1}$]). For time spent sitting, a significantly greater reduction from the pre-assessment values (469 mins) were observed at the post-assessment for the IPC group (Mean ± SD [95%CI]; 385 ± 89 mins [340–432 mins]) compared to CON (Mean ± SD [95%CI]; 474± 89 mins [423–525 mins]).

## Additional measures

Correlation analysis showed the change in daily average IPC wear time from the first to last week of the intervention was weakly correlated with peripheral SBP (*rho* = 0.11) and cSBP (*rho* = 0.13). Weak to moderate correlations were found between changes in daily average weekly physical activity and both SBP (*rho* = 0.11) and cSBP (*rho* = 0.26). The change in 6MWT distance was weakly correlated with IPC wear time (rho = -0.20) and the change in weekly physical activity (rho = 0.25, p > 0.05). Additionally, a weak to moderate,

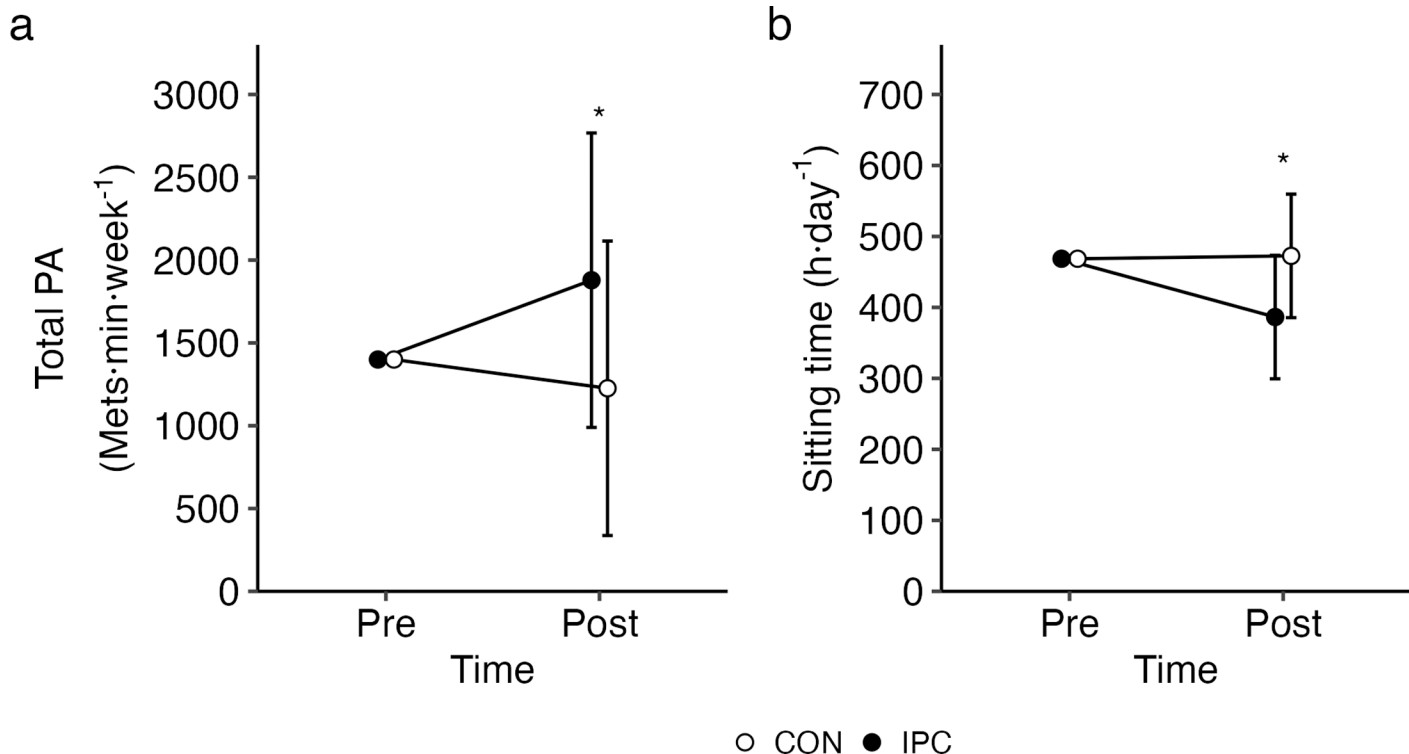

**Fig 5. Mean total physical activity (5A) and mean sitting time (5B) for IPC and CON at pre- and post-intervention assessments.**

non-significant correlation was observed between the total number of IPC sessions (range; 20 to 77) and the change in weekly physical activity (r = 0.26, p > 0.05).

## Discussion

This study assessed the effect of using a home-based intermittent pneumatic compression (IPC) device (GMOVE Suit) on functional and vascular outcomes in people with chronic stroke. Our study demonstrated that 12 weeks of home-based IPC training significantly increased 6MWT distance, and improved (reduced) peripheral SBP and cSBP, in those individuals randomized to the IPC condition. These observed improvements, coupled with increased perceived physical activity and reduced sedentary behavior, are pertinent outcomes that support the use of IPC training for "at-home" rehabilitation therapy in individuals living with stroke.

Enhancing walking ability is one of the most commonly sought goals in stroke rehabilitation and is directly linked to improvements in post-stroke quality of life [40]. In our study, we demonstrated a 12% improvement in 6MWT walking distance between pre- and post-intervention assessments for the IPC group. This is encouraging as the minimum clinical difference for a change in the 6MWT is ≥ 13% [24]. To complement this finding, we observed a ~ 15% increase in IPC device wear time and nearly a three unit increase in RPE when wearing the IPC at the end of the 12-week program. The "at-home" nature of our study and device accessibility may have encouraged participants to undertake a higher volume of walking compared to the control group, as the participants could wear the IPC device at any time or day during the program period. Nevertheless, our study demonstrated weak correlations when comparing the change in IPC wear time and the change in weekly physical activity with changes in 6MWT performance. Despite this, it must be noted that the favorable changes observed for the 6MWT were not observed for

the TUG or for the sit-to-stand functional outcome measures. The TUG test primarily evaluates mobility, balance and the functional movements of standing-up, walking a short distance, turning and sitting down. Although the IPC intervention included movements that overlap with the tasks assessed in the TUG test, the primary focus was on using the IPC device during regular walking activities. It did not specifically target balance or the complex movement transitions, such as standing up, turning, and sitting down, that are key components of the TUG test. This may explain why no significant improvements were observed in the TUG test [41].

Despite the greater accessibility and exercise dosage afforded by home-based over-ground IPC training programs for chronic stroke patients, there were no significant changes in BBS or Fugl Meyer (upper and lower-extremity scores) for either condition. This may again relate to the fact that walking was the primary exercise activity in the intervention, with balance-type activities of secondary importance. Nevertheless, both conditions did report a perceived improvement in their balance confidence as determined by the ABC scale, an indicator of an individual's fear of falling. The 8% improvement observed in balance confidence for the IPC group was greater than that seen for the control group. In comparison to recent research with subacute stroke patients (42 ± 17 days post-stroke), our observed improvements with a chronic stroke population were smaller than those reported following a 2.5-week (12 training session) perturbation-based balance training program (mean improvement of 18%), but better than a 2.5 week (12 training session) weight shifting and gait training program (mean improvement of 3%) [42]. A potential explanation of the positive effect of primarily walking-focused programs on the fear of falling may be related to the nature of the ABC Scale, in which most items involve the components of ambulation or transfer [43]. These factors may combine to explain why exercise that focuses on improving functional mobility post-stroke, such as our study, leads to a therapeutic effect on the fear of falling [44]. To complement this, it is also important to report that there were no AE's associated with falling in this study.

Previous studies have shown that aerobic training interventions can reduce SBP by up to 5 mmHg [45], and that such a change through pharmacological management can lower the risk of major cardiovascular events by around 10% [46]. Thus, the ~7 mmHg reduction in blood pressure observed in IPC participants aligns with previous research and is especially promising, given the limited mobility of this population and the low-intensity, primarily walking-based nature of the IPC intervention. However, our study did demonstrate a large increase in physical activity (~37%) and reduced sedentary behavior (18%) on completion of the 12-week program for those using the IPC device. For the IPC group, these changes in lifestyle behaviors may have been facilitated, in part, by the access to the IPC device, with participants increasing their average weekly usage over the course of the intervention period (~4 to 5 days). If IPC devices have the potential to increase physical activity, they may be considered a crucial and effective intervention for lowering blood pressure in individuals who have experienced a stroke [47].

Despite these encouraging changes in blood pressure, there were no other statistical changes in other vascular outcomes. Carotid-femoral pulse wave velocity (cfPWV), for example, is a strong predictor of CVD in a variety of clinical populations [48]. In our study, an average decrease of 0.35 m/s in cfPWV was observed in the IPC group at the post-assessment, compared to a 0.08 m/s increase in the control group. As a 1 m/s reduction in PWV is the minimal clinically important difference and is strongly associated with decreased CVD risk [48], our lower-intensity IPC program may not be as effective in impacting upon regional arterial stiffness. Since AIx is a derivative of PWV, it is unsurprising that there were no differences between groups for such outcomes. Previous research has consistently demonstrated favorable changes in vascular outcomes (e.g., cfPWV) when moderate to vigorous volumes of physical activity are prescribed in training interventions [49,50]. Higher-intensity exercise

increases shear stress on the arterial walls, stimulating the release of nitric oxide from the endothelium. This promotes vasodilation, improves arterial compliance, and reduces arterial stiffness, as reflected in a decrease in PWV [49]. Lower-intensity exercise, such as our IPC intervention, may not initiate such physiological adaptations. However, they may promote better adherence and be more attainable, as enjoyment and well-being are strong motivators for long-term engagement [51,52]. Given our encouraging findings regarding the potential use of this technology and the implementation of the IPC program to aid the recovery of people living with stroke over the longer-term, it is important to monitor measures of enjoyment in both the short- and longer-term.

It is important to contextualize the limitations and strengths of this research. Firstly, there was evidence of sampling bias, as the recruited sample may not represent a typical stroke population. Participants were recruited from a private neuro-physiotherapy practice, meaning they were likely highly motivated to participate in rehabilitation due to the financial investment associated with private physiotherapy services. This ultimately limits the generalizability of the study findings. Additionally, the study's external validity is affected because highly motivated individuals are more likely to fully engage with and adhere to study protocols. This higher level of compliance could make the intervention seem more effective than it actually would be in a broader stroke population, potentially resulting in an overestimation of its true efficacy. Furthermore, and similar to the vast majority of rehabilitation studies, performance bias could be evident in the study due to participants and therapists not being blinded to group allocation [53]. It is also worth noting that the recruited population exhibited a diverse range of Functional Ambulation Classifications, thereby increasing the study heterogeneity and reducing the possibility of seeing statistical differences in certain study outcomes (e.g., TUG, 10MWT). A further limitation is that participants did not strictly adhere to the designated RPE throughout the 12-week intervention. For example, on completion of the program, the reported RPE of ~ 15 (hard feeling of exertion) was lower than that prescribed (RPE 17; Very hard perception of exertion). Further research is needed to explore whether this was due to participant motivation or the IPC device being unable to help participants to exercise at higher exercise intensities. If the latter, further research is needed to determine whether alternative parameters (e.g., heart rate) could be a better tool to progress participants, whether varying pressures and IPC device dosage elicit more favorable effects on patients, and whether the technology is more suited for acute or sub-acute stroke patients, or bilateral populations (e.g., multiple sclerosis, spinal cord injury). Considering the sampling bias identified in this study, future research should also evaluate the generalizability of the observed positive effects of IPC therapy to a broader stroke population. Furthermore, investigations could examine whether stratification by stroke severity (e.g., minor, moderate, severe) yields greater improvements in functional mobility and cardiovascular health, while additional studies are warranted to assess the potential synergistic effects of integrating IPC therapy with structured exercise interventions, e.g., 8- to 12-week aerobic training program.

A key aspect of successful behavior change is that individuals maintain their lifestyle modifications even after the initial motivation or stimulus is no longer present. As such, a limitation of the present study is that we are unable to determine how long the observed benefits in 6MWT and blood were observed once the experimental manipulation of the IPC device was removed. Previous research with chronic stroke patients that have used other forms of lower limb technology [13] and that showed a similar change in 6MWT performance on completion of a 10-week walking-based program demonstrated that the improvements reported in 6MWT post-intervention were maintained for at least 3 months after cessation of the program. If the long-term benefits of an IPC program are sustained in terms of functional performance and vascular health, it could result in significant enhancements in mental health, increased capacity

for activities of daily living, and ultimately, an overall improvement in the individual's quality of life. Despite these limitations, there are many strengths and practical implications associated with the study. For example, the use of a randomized controlled trial, undertaking participants' pre- and post-intervention assessments in a controlled environment whereby differences in baseline values were accounted for, and the ecological validity of implementing IPC device technology within a home-based environment, are all strengths of the study.

In conclusion, our study demonstrated improvements in 6MWT and systolic blood pressure following 12 weeks of home-based IPC training. These changes, in combination with an increase in physical activity and reduced sedentary behaviors, are important positive findings when considering the use of IPC training for "at-home" rehabilitation therapy for people with stroke. Further research is needed to explore the impact of the volume (frequency, duration etc.) of the IPC exposure on functional and vascular health outcomes, to determine whether the observed benefits are maintained for a longer period (e.g., 3- or 6-month follow-up), whether home-based IPC therapy could serve as an adjunct to conventional rehabilitation, and if similar findings are observed with other neurological populations.

## Supporting information

**S1 Table. Functional outcomes and SF-12 (physical and mental component) scores for IPC (n = 15) and CON (n = 16) groups at pre- and post-intervention.**
Data reported as mean (± SD) or as a percentage (%).

**S2 Table. Mean (± SD) vascular outcomes from IPC (n = 15) and CON (n = 16) groups at pre- and post-intervention.**
(DOCX)

**S1 Appendix. C_Research Protocol.**
(DOCX)

**S1 Data. CONSORT Checklist.**
(DOCX)

**S2 Data. IPC_Data_Plos One.**
(XLSX)

## Acknowledgements

The authors would like to thank all the people who participated in the trial and their families.

## Author contributions

**Conceptualization:** James Faulkner, Amy Dennis-Jones, Louis Martinelli, Helen Hobbs.

**Data curation:** James Faulkner, Eloise Paine, Nick Hudson.

**Formal analysis:** James Faulkner, Eloise Paine, Scott Hannah.

**Funding acquisition:** James Faulkner.

**Investigation:** Eloise Paine, Nick Hudson, Scott Hannah, Amy Dennis-Jones, Louis Martinelli.

**Methodology:** Eloise Paine, Amy Dennis-Jones, Louis Martinelli.

**Supervision:** James Faulkner.

**Writing – original draft:** James Faulkner, Eloise Paine.

**Writing – review & editing:** James Faulkner, Eloise Paine, Nick Hudson, Scott Hannah, Amy Dennis-Jones, Louis Martinelli, Helen Hobbs.

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
