## [Decision Letter · Decision Letter 0]

10 Jul 2024

PONE-D-24-13580Effect of home-based dynamic intermittent pneumatic compression therapy on vascular and functional health outcomes in chronic stroke: A randomized controlled clinical trialPLOS ONE

Dear Dr. Faulkner,

Thank you for submitting your manuscript to PLOS ONE. After careful consideration, we feel that it has merit but does not fully meet PLOS ONE’s publication criteria as it currently stands. Therefore, we invite you to submit a revised version of the manuscript that addresses the points raised during the review process.

**ACADEMIC EDITOR: ** All issues raised by reviewers are required. The present manuscript is affected by several shortcomings that need to be addressed poit by point in rder to better support the conclusion. The discussion should be reshaped and limitations of the study should be better addressed.

We look forward to receiving your revised manuscript.

Kind regards,

Vincenzo Lionetti, M.D., PhD

Academic Editor

PLOS ONE

Journal Requirements:

"JF Received funding for the research

Grant number: N/A

The full name of each funder: WinBack Medical

URl: https://winbackmedical.com/

Did the sponsors or funders play any role in the study design, data collection and

analysis, decision to publish, or preparation of the manuscript?: NO"

3. In this instance it seems there may be acceptable restrictions in place that prevent the public sharing of your minimal data. However, in line with our goal of ensuring long-term data availability to all interested researchers, PLOS’ Data Policy states that authors cannot be the sole named individuals responsible for ensuring data access (http://journals.plos.org/plosone/s/data-availability#loc-acceptable-data-sharing-methods).

Reviewers' comments:

Reviewer's Responses to Questions

**Comments to the Author**

1. Is the manuscript technically sound, and do the data support the conclusions?

Reviewer #1: Partly

Reviewer #2: Yes

Reviewer #3: Yes

2. Has the statistical analysis been performed appropriately and rigorously? 

Reviewer #1: No

Reviewer #2: No

Reviewer #3: Yes

3. Have the authors made all data underlying the findings in their manuscript fully available?

Reviewer #1: No

Reviewer #2: Yes

Reviewer #3: Yes

4. Is the manuscript presented in an intelligible fashion and written in standard English?

Reviewer #1: Yes

Reviewer #2: Yes

Reviewer #3: Yes

5. Review Comments to the Author

Reviewer #1: The manuscript is well-written and follows a logical structure. The study addresses an important issue in stroke rehabilitation and provides evidence supporting the use of home-based IPC therapy. Major revisions are needed for clarity and completeness.

1. Abstract: Please add a sentence on the significance and clinical implications of the findings.

2. Page 6, lines 135-138: The authors need to elaborate and explain the rationale behind reporting some parameters here and others in a separate manuscript.

3. Page 7: Provide more detail on the randomisation process, including blocks, strata, etc.

4. Page 7: Intervention: It is unclear whether the intervention is only IPC or IPC plus exercise. Please could the authors clarify the exact nature of the intervention?

5. Follow-up: Ideally, this should be mentioned in a separate section.

6. Page 9: Functional outcome measures: Who measured these? Were they blinded?

7. Page 12: Data analysis: Provide more explanation on the use of the independent samples t-test and where it was applied. If I understand correctly, it is used to compare follow-up differences between the functional outcome parameters. If so, would it be paired-t test? Also, I would expect to see comment on the distribution of the data.

8. Page 12, lines 288-289: Is there missing data generation? This needs to be clearly explained.

9. Methods: What adverse events were measured and how?

10. Methods: Which parameters were primary and which were secondary outcomes?

11. Methods: Was a sample size calculation performed?

12. Results: The authors need to clarify whether IPC or the exercise affected the findings. They should provide a correlation between IPC time and blood pressure and 6MWT, as well as exercise time and blood pressure and 6MWT.

13. If it is exercise rather than IPC explains the difference, do they need to update the title accordingly?

14. Discussion: Expand on future research directions and the potential long-term impacts of the findings.

15. Table 3: Remove the page numbers from the last column.

16. Figures 5 & 6: Are these error plots? If so, where is the confidence interval (CI) of the baseline?

Reviewer #2: This is an interesting study and commend the researchers.

Consider restructuring sessions, i.e details re inclusion/exclusion criteria should come prior to methods.

Some comments:

Line 92-93, make it clear which is the primary (Functional measured using 6MWT), secondary objective. Because when you list the outcomes, looks like the vascular is first, however sample size is based on functional outcome.

Elements of randomisation approach is missing, i.e line 273. i.e what is the allocation ratio - mention this here

Data analysis - as this an individualised RCT, not recommended to test baseline factors, any differences observed are more likely due to chance - recommend revising this.

Recommend to move the sentence, re reporting results in accordance to CONSORT under data analysis.

Define intention to treat, i.e analysed as they were randomised regardless of which intervention they actually received

Repeated measures, is this relating to outcomes collected at 1 and 12 weeks- if so make this clear, i.e all tables will need revision in terms of information presented, i.e Pre=Baseline, Post = which time point(s)?

Sentence in line 297 - studies are not terminated due reaching sample size, this would be defined as end of study, stating termination implies prematurely stopped - which I don't think is the case.

Line 323 - does not make sense, i.e. no differences yet p<0.05 is reported and table 3 does shows some differences.

Table 1, add the in (n=??) in the table itself next to IPC, Control respectively.

Add percentages for male, female. Add range for age in both groups.

Rationale as to why safety outcomes were not measured, if at all?

CONSORT diagram, study is repeated, is there data to show what time point people dropped off?

Reviewer #3: This manuscript addresses a significant issue in stroke rehabilitation and presents promising findings on the use of home-based intermittent pneumatic compression therapy. Overall, the paper is well structured and the analysis is well conducted. Please find below some specific comments.

The study design is robust, using a randomized controlled trial format and adhering to CONSORT guidelines, and with a detailed descriptions of the inclusion and exclusion criteria, randomization process, and intervention protocols. On the other hand, the sample size calculation rationale is mentioned but lacks detail on the expected effect size used in the calculation. There is also a potential bias in participant selection, as participants were recruited from a private neuro-physiotherapy practice, which may not represent the general stroke population. Finally, blinding is appropriately addressed, but the lack of blinding in participants and therapists could introduce performance bias. More emphasis on this limitation and how it was mitigated would be beneficial.

The hypothesis is clearly stated, but a brief mention of the primary outcomes expected would strengthen the rationale.

The results section could be improved by providing more detailed statistical values, including confidence intervals for the primary outcomes.

The discussion of non-significant findings is brief. Providing a more in-depth analysis of why certain expected outcomes were not achieved would add value.

The discussion of limitations is somewhat superficial. More depth is needed in discussing the potential impact of the study’s limitations on the results and the generalizability of the findings.

In the conclusions, the call for further research is appropriate but could be more specific about the next steps and potential studies to build on these findings.

The abstract succinctly summarizes the background, methods, results, and conclusions. However, it could be more concise by eliminating some redundant phrases. Additionally, the abstract mentions significant improvements in specific health outcomes but lacks specificity on the statistical values, which would help readers quickly grasp the extent of the findings.

Some sections of the introduction could be more concise. For example, the paragraph discussing the general benefits of physical activity in stroke rehabilitation could be shortened, as it reiterates well-known information.

The manuscript is globally well-written but would benefit from a thorough proofreading to correct minor grammatical errors and improve readability.

6. PLOS authors have the option to publish the peer review history of their article (what does this mean? ). If published, this will include your full peer review and any attached files.

**Do you want your identity to be public for this peer review?** For information about this choice, including consent withdrawal, please see our Privacy Policy .

Reviewer #1: **Yes: ** kausik Chatterjee

Reviewer #2: No

Reviewer #3: No

---

## [Author Response · Author response to Decision Letter 1]

13 Sep 2024

PONE-D-24-13580

Effect of home-based dynamic intermittent pneumatic compression therapy on vascular and functional health outcomes in chronic stroke: A randomized controlled clinical trial

PLOS ONE

Dear Dr. Faulkner,

Thank you for submitting your manuscript to PLOS ONE. After careful consideration, we feel that it has merit but does not fully meet PLOS ONE’s publication criteria as it currently stands. Therefore, we invite you to submit a revised version of the manuscript that addresses the points raised during the review process.

Thank you for your response and for allowing us to revise the manuscript. The three reviewers have provided a series of insightful comments which have led to amendments to the paper. As a result, we feel the paper has been strengthened and makes a novel contribution to the literature. We look forward to receiving further feedback following this re-submission

ACADEMIC EDITOR: All issues raised by reviewers are required. The present manuscript is affected by several shortcomings that need to be addressed point by point in order to better support the conclusion. The discussion should be reshaped and limitations of the study should be better addressed.

The manuscript has been amended in light of the reviewer and editor suggestions.

Journal Requirements:

The manuscript has been checked throughout in accordance with PLOS ONE’s style requirements.

"JF Received funding for the research

Grant number: N/A

The full name of each funder: WinBack Medical

URl: https://winbackmedical.com/

Did the sponsors or funders play any role in the study design, data collection and

analysis, decision to publish, or preparation of the manuscript?: NO"

We have amended this statement as advised and have updated the cover letter.

3. In this instance it seems there may be acceptable restrictions in place that prevent the public sharing of your minimal data. However, in line with our goal of ensuring long-term data availability to all interested researchers, PLOS’ Data Policy states that authors cannot be the sole named individuals responsible for ensuring data access (http://journals.plos.org/plosone/s/data-availability#loc-acceptable-data-sharing-methods).

A non-author contact has been included as advised.

Non-author contact includes: Dr Samantha Scallan, Chair Of University Ethics Committee, ethics1@winchester.ac.uk.

Long-term data storage will be ensured by storing data with the Department of Sport, Exercise and Health’s Onedrive site.

Ethics statement is only included in the methods and has been removed from other locations within the manuscript (e.g., before the references)

Information pertaining to the Supplementary Table (now entitled S1 and S2 Table in accordance with guidelines) has been included at the end of the manuscript as requested.

5. Review Comments to the Author

Reviewer #1: The manuscript is well-written and follows a logical structure. The study addresses an important issue in stroke rehabilitation and provides evidence supporting the use of home-based IPC therapy. Major revisions are needed for clarity and completeness.

Thank you for your positive comments and for providing diligent feedback. We have amended the manuscript in light of the observations you have made.

1. Abstract: Please add a sentence on the significance and clinical implications of the findings.

Thank you for the comment. In light of this observation we have adjusted the conclusion to now read:

‘The observed improvements in blood pressure and 6MWT, in combination with an increase in physical activity and reduced sedentary behaviors, all pertinent characteristics of cardiovascular and functional health in stroke patients, are important positive findings when considering the use of IPC training as a clinical training aid for “at home” rehabilitation therapy for chronic stroke survivors.’

2. Page 6, lines 135-138: The authors need to elaborate and explain the rationale behind reporting some parameters here and others in a separate manuscript.

Additional information has been included to provide further clarity regarding this decision. It now reads:

‘Other measures of vascular health were also captured, including brachial-femoral and femoral-anterior tibial PWV, however they will be reported in a separate manuscript as they are exploring some methodological considerations associated with assessing PWV in comparison to the widely used and cited cfPWV.’

3. Page 7: Provide more detail on the randomisation process, including blocks, strata, etc.

We have amended the content to provide some specific information on our randomization process.

‘Block randomization (2x IPC participants, 2x Control participants per block) was used to ensure an equal balance of participants between groups. The order within each block was randomized.‘

4. Page 7: Intervention: It is unclear whether the intervention is only IPC or IPC plus exercise. Please could the authors clarify the exact nature of the intervention?

The intervention was only IPC. Both IPC and control participants were advised to engage in a minimum of 30 minutes of habitual daily physical activity. During this time, IPC participants were advised to wear the IPC device.

5. Follow-up: Ideally, this should be mentioned in a separate section.

We have inserted the following content in recognition of the follow-up assessment.

‘Follow-up

All outcome assessments which were undertaken within the pre-assessments were repeated at follow-up (post-assessments).’

6. Page 9: Functional outcome measures: Who measured these? Were they blinded?

At the end of the experimental design we state that ‘outcome assessors and data analysts were blinded from group allocation’. Physical therapists with experience in administering the functional outcome measures undertook the functional assessments. We have included information pertaining to this at the start of the functional outcome section. It now reads:

‘All functional outcome measures were undertaken by physical trainers who had > 10 years of experience in administering these assessments to neurological populations.’

7. Page 12: Data analysis: Provide more explanation on the use of the independent samples t-test and where it was applied. If I understand correctly, it is used to compare follow-up differences between the functional outcome parameters. If so, would it be paired-t test? Also, I would expect to see comment on the distribution of the data.

Independent samples t-tests were used to compare baseline outcome data (e.g., participant age, time since stroke, baseline FAC) between those randomised to the IPC and control group. As such, this is the correct test to use as a paired samples t-test is more appropriate if looking at changes in outcomes between time points (pre- to post for example). With regards to the assessment of data distribution we do have the following statement that relates to this

‘To assess the effect of the IPC therapy on functional, vascular, and quality of life health outcomes, and following checks for normality to determine the data were parametric,…’. However, to clarify the measures used to assess the distribution of the data (e.g., parmetric or non-parametric) we have included the following:

‘… following checks for normality to determine the data were parametric (e.g., skewness, kurtosis, Shapiro-Wilk test of normality, histograms),…’

8. Page 12, lines 288-289: Is there missing data generation? This needs to be clearly explained.

We feel this is explained, however, we have adapted our sentence to help clarify the information:

‘An intention-to-treat analysis was applied to all repeated-measures procedures, with participants analyzed as randomized, using the last available data to replace any subsequent missing assessments.’

9. Methods: What adverse events were measured and how?

Recognition of the reporting adverse events are now included in the methods and results.

The method section includes the following:

‘Study-related and non-study-related serious adverse event (SAE) and adverse event (AE) tracking occurred throughout the study. SAE and AEs were recorded in the exercise diary and were discussed with therapists during their face-to-face or telephone-based discussions.’

The results section now includes the following

‘One IPC participant terminated their involvement in the study four weeks after study allocation due to a lack of engagement with the IPC device. There were no “probably study-related” SAEs, but there was one “non-study related” SAE (seizure). There were 20 total study-related AEs which included 15 reported pressure discomforts, two minor skin abrasions and three musculoskeletal concerns (e.g., muscle tightness after IPC intervention). All study-related AEs were resolved, and participants continued in the study. There were no falls during the IPC intervention.’

10. Methods: Which parameters were primary and which were secondary outcomes?

The primary outcome measure to the study was the 6MWT. This is why the 6MWT was used as the outcome of interest in the sample size calculation. Further recognition of this has been included in the manuscript within the functional outcome measures section:

‘The 6MWT was the primary outcome measure to the study and provides…’

Accordingly, we have changed the structure of the methods, results and discussion to ensure the primary outcome is discussed first. All other outcomes (e.g., vascular outcomes, TUG, physical activity etc.) are considered secondary outcomes.

11. Methods: Was a sample size calculation performed?

Yes. This is presented at the bottom of the opening paragraph to the methods section.

12. Results: The authors need to clarify whether IPC or the exercise affected the findings. They should provide a correlation between IPC time and blood pressure and 6MWT, as well as exercise time and blood pressure and 6MWT.

Thank you for the comment. We have now included recognition of the above correlations in both the data analysis and results sections to the manuscript to provide some additional context. For example, we have included the following content in the data analysis:

‘Pearson correlation coefficients and Spearman’s rho were used to explore whether: i. changes in average daily IPC wear time over the course of the IPC intervention, and ii. changes in total physical activity as determined by the IPAQ, were correlated with changes in the 6MWT and pertinent blood pressure markers’

13. If it is exercise rather than IPC explains the difference, do they need to update the title accordingly?

Thank you for your comment, It not necessarily exercise that is the focus of the study, but it is the use of IPC during habitual activity. As such, we have adjusted the comment to reflect this observation from the reviewer. It now reads:

‘Effect of using home-based dynamic intermittent pneumatic compression therapy during habitual physical activity on vascular and functional health outcomes in chronic stroke: A randomized controlled clinical trial.’

14. Discussion: Expand on future research directions and the potential long-term impacts of the findings.

We feel that our discussion highlights a number of pertinent future research directions and considerations as we state the following:

‘Further research is needed to explore whether this was due to participant motivation or the IPC device being unable to help participants to exercise at higher exercise intensities. If the latter, further research is needed to determine whether alternative parameters (e.g., heart rate) could be a better tool to progress participants, whether varying pressures and IPC device dosage elicit more favorable effects on patients, and whether the technology is more suited for acute or sub-acute stroke patients, or bilateral populations (e.g., multiple sclerosis, spinal cord injury).’

We have also included in the discussion the following sentence to further highlight the potential long-term importance of such innovation:

‘If the long-term benefits of an IPC program are sustained in terms of functional performance and vascular health, it could result in significant enhancements in mental health, increased capacity for activities of daily living, and ultimately, an overall improvement in the individual's quality of life.’

15. Table 3: Remove the page numbers from the last column.

We can not see any page numbers in the last column.

16. Figures 5 & 6: Are these error plots? If so, where is the confidence interval (CI) of the baseline?

These are not error plots. This is data from the 6MWT and physical activity/sitting time data following Analysis of covariance (ANCOVA). As we wanted to control for any baseline (pre) differences between IPC and Con groups for these outcomes, we included these pre-data outcomes as the covariate. As such, when using ANCOVA, only SD’s are reported for the post-data. We have included the key data associated with Figures 4 and 5 within the results section, including 95% CI’s and effect size (partial eta squared) where necessary.

Reviewer #2: This is an interesting study and commend the researchers.

Thank you for your positive feedback on the manuscript.

Consider restructuring sessions, i.e details re inclusion/exclusion criteria should come prior to methods.

As we have followed the guidance of the journal and the CONSORT checklist, we are content with the structure of the methods. For example, the CON

---

## [Decision Letter · Decision Letter 1]

7 Oct 2024

PONE-D-24-13580R1Effect of using home-based dynamic intermittent pneumatic compression therapy during habitual physical activity on functional and vascular health outcomes in chronic stroke: A randomized controlled clinical trialPLOS ONE

Dear Dr. Faulkner,

Thank you for submitting your manuscript to PLOS ONE. After careful consideration, we feel that it has merit but does not fully meet PLOS ONE’s publication criteria as it currently stands. Therefore, we invite you to submit a revised version of the manuscript that addresses the points raised during the review process.

**ACADEMIC EDITOR: ** Some issues raised by one expert reviewer remain. Please provide an accurate response.

We look forward to receiving your revised manuscript.

Kind regards,

Vincenzo Lionetti, M.D., PhD

Academic Editor

PLOS ONE

Journal Requirements:

Reviewers' comments:

Reviewer's Responses to Questions

**Comments to the Author**

1. If the authors have adequately addressed your comments raised in a previous round of review and you feel that this manuscript is now acceptable for publication, you may indicate that here to bypass the “Comments to the Author” section, enter your conflict of interest statement in the “Confidential to Editor” section, and submit your "Accept" recommendation.

Reviewer #1: (No Response)

Reviewer #2: All comments have been addressed

Reviewer #3: All comments have been addressed

2. Is the manuscript technically sound, and do the data support the conclusions?

Reviewer #1: Yes

Reviewer #2: Yes

Reviewer #3: Yes

3. Has the statistical analysis been performed appropriately and rigorously? 

Reviewer #1: Yes

Reviewer #2: Yes

Reviewer #3: Yes

4. Have the authors made all data underlying the findings in their manuscript fully available?

Reviewer #1: Yes

Reviewer #2: Yes

Reviewer #3: Yes

5. Is the manuscript presented in an intelligible fashion and written in standard English?

Reviewer #1: Yes

Reviewer #2: Yes

Reviewer #3: Yes

6. Review Comments to the Author

Reviewer #1: Although the authors have addressed most of the points generated during the review, please

• Refine the abstract to better highlight the broader clinical implications of IPC therapy for stroke rehabilitation.

• Provide additional detail on the randomisation process (clearer explanation of the block sizes and strata) and handling of missing data in the methodology.

• Clarify the integration of IPC with habitual physical activity to avoid ambiguity in the intervention description and explain how this integration differs from traditional exercise interventions.

• Expand the discussion of non-significant findings and provide hypotheses for why certain outcomes did not improve.

• Acknowledge the potential selection bias in more detail, particularly the limitations of recruiting highly motivated participants from a private clinic. Discuss how this may affect the external validity of the findings.

• Specify future research directions more clearly, by providing more concrete suggestions for future studies, such as exploring different stroke severity levels or testing IPC therapy in combination with more structured exercise regimens.

Reviewer #2: (No Response)

Reviewer #3: The Authors have revised rather extensively their manuscript according to my comments and those by the other Reviewers. I have no further comments.

7. PLOS authors have the option to publish the peer review history of their article (what does this mean? ). If published, this will include your full peer review and any attached files.

**Do you want your identity to be public for this peer review?** For information about this choice, including consent withdrawal, please see our Privacy Policy .

Reviewer #1: **Yes: ** Kausik Chatterjee

Reviewer #2: No

Reviewer #3: No

---

## [Author Response · Author response to Decision Letter 2]

7 Jan 2025

Review Comments to the Author

Reviewer #1: Although the authors have addressed most of the points generated during the review, please

• Refine the abstract to better highlight the broader clinical implications of IPC therapy for stroke rehabilitation.

We have adjusted the conclusion of the abstract to better reflect the broader clinical implications as advised. It now reads:

‘The observed improvements in functional mobility, cardiovascular health, increased physical activity and reduced sedentary time demonstrates important clinical implications of ‘home-based’ IPC therapy as a clinical training aid for stroke rehabilitation. Home-based IPC therapy could serve as an adjunct to conventional rehabilitation, however, further research is needed to determine whether IPC therapy can sustain or improve function over time for individuals in the chronic stage of recovery.’

• Provide additional detail on the randomisation process (clearer explanation of the block sizes and strata) and handling of missing data in the methodology.

To improve the clarity the following information has been adapted regarding information on the Block randomization process:

‘Block randomization was employed to ensure an equal distribution of participants across the groups. The study used eight blocks, each consisting of four participants, two assigned to the IPC group and two to the Control group, to meet the required sample size. Within each block, participants were randomized to either the IPC or control group.’

In terms of the handling of missing data, we are unsure what further information is required as we feel that the information provided clearly articulates the process. On page 14 under the Data analysis subheading we stated:

‘An intention-to-treat analysis was applied to all repeated-measures procedures, with participants analyzed as randomized, using the last available data to replace any subsequent missing assessments’

• Clarify the integration of IPC with habitual physical activity to avoid ambiguity in the intervention description and explain how this integration differs from traditional exercise interventions.

Thank you for your comments. We have removed the term “habitual” from the manuscript when referring to the IPC intervention, as it does not accurately reflect the prescribed program. Participants were instructed to wear the IPC device for at least 30 minutes daily, during which they were encouraged to walk and engage in activities such as step-ups and sit-to-stands at prescribed RPEs. While the program shares some similarities with a traditional exercise program, its intensity is likely lower than that of a gym-based aerobic or resistance training program, due to the focus on walking and the specific needs of the population group.

• Expand the discussion of non-significant findings and provide hypotheses for why certain outcomes did not improve.

We have provided additional context as to why changes in the TUG were not observed. We have adapted the second paragraph to the discussion to now read:

‘The TUG test primarily evaluates mobility, balance and the functional movements of standing-up, walking a short distance, turning and sitting down. Although the IPC intervention included movements that overlap with the tasks assessed in the TUG test, the primary focus was on using the IPC device during regular walking activities. It did not specifically target balance or the complex movement transitions, such as standing up, turning, and sitting down, that are key components of the TUG test. This may explain why no significant improvements were observed in the TUG test’.

However, when discussing the Fugl Meyer and BBS results, we do not think any further information is needed due to the following:

‘…, there were no significant changes in BBS or Fugl Meyer (upper and lower-extremity scores) for either condition. This may again relate to the fact that walking was the primary exercise activity in the intervention, with balance-type activities of secondary importance.’

Finally, when commenting on the non-significant findings related to cfPWV and AIX, the explanation around higher-intensity exercise has been expanded to now read:

‘As a 1 m/s reduction in PWV is the minimal clinically important difference and is strongly associated with decreased CVD risk [48], our lower-intensity IPC program may not be as effective in impacting upon regional arterial stiffness. Since AIx is a derivative of PWV, it is unsurprising that there were no differences between groups for such outcomes. Previous research has consistently demonstrated favorable changes in vascular outcomes (e.g., cfPWV) when moderate to vigorous volumes of physical activity are prescribed in training interventions [49, 50]. Higher-intensity exercise increases shear stress on the arterial walls, stimulating the release of nitric oxide from the endothelium. This promotes vasodilation, improves arterial compliance, and reduces arterial stiffness, as reflected in a decrease in PWV [49]. Lower-intensity exercise, such as our IPC intervention, may not initiate such physiological adaptations. However, they may promote better adherence and be more attainable, as enjoyment and well-being are strong motivators for long-term engagement [51] [52].’

• Acknowledge the potential selection bias in more detail, particularly the limitations of recruiting highly motivated participants from a private clinic. Discuss how this may affect the external validity of the findings.

We have restructured this section to the manuscript so that the limitations associated with the sample are clearly articulated. It now reads:

‘Firstly, there was evidence of sampling bias, as the recruited sample may not represent a typical stroke population. Participants were recruited from a private neuro-physiotherapy practice, meaning they were likely highly motivated to participate in rehabilitation due to the financial investment associated with private physiotherapy services. This ultimately limits the generalizability of the study findings. Additionally, the study's external validity is affected because highly motivated individuals are more likely to fully engage with and adhere to study protocols. This higher level of compliance could make the intervention seem more effective than it actually would be in a broader stroke population, potentially resulting in an overestimation of its true efficacy.’

• Specify future research directions more clearly, by providing more concrete suggestions for future studies, such as exploring different stroke severity levels or testing IPC therapy in combination with more structured exercise regimens.

We have revised the future research section to provide a clearer outline of potential directions for future work. It now reads:

‘If the latter, further research is needed to determine whether alternative parameters (e.g., heart rate) could be a better tool to progress participants, whether varying pressures and IPC device dosage elicit more favorable effects on patients, and whether the technology is more suited for acute or sub-acute stroke patients, or bilateral populations (e.g., multiple sclerosis, spinal cord injury). Considering the sampling bias identified in this study, future research should also evaluate the generalizability of the observed positive effects of IPC therapy to a broader stroke population. Furthermore, investigations could examine whether stratification by stroke severity (e.g., minor, moderate, severe) yields greater improvements in functional mobility and cardiovascular health, while additional studies are warranted to assess the potential synergistic effects of integrating IPC therapy with structured exercise interventions, e.g., 8- to 12-week aerobic training program.’

Reviewer #2: (No Response)

Reviewer #3: The Authors have revised rather extensively their manuscript according to my comments and those by the other Reviewers. I have no further comments.

Thank you for reading the revised manuscript and for your engagement in the process.

---

## [Editor Report · Decision Letter 2]

24 Jan 2025

Effect of using home-based dynamic intermittent pneumatic compression therapy during periods of physical activity on functional and vascular health outcomes in chronic stroke: A randomized controlled clinical trial

PONE-D-24-13580R2

Dear Dr. Faulkner,

We’re pleased to inform you that your manuscript has been judged scientifically suitable for publication and will be formally accepted for publication once it meets all outstanding technical requirements.

Kind regards,

Vincenzo Lionetti, M.D., PhD

Academic Editor

PLOS ONE
---

## [Editor Report · Acceptance letter]

PONE-D-24-13580R2

PLOS ONE

Dear Dr. Faulkner,

I'm pleased to inform you that your manuscript has been deemed suitable for publication in PLOS ONE. Congratulations! Your manuscript is now being handed over to our production team.

Kind regards,

on behalf of

Prof. Vincenzo Lionetti

Academic Editor

PLOS ONE